# Toxicity Assessment of Octachlorostyrene in Human Liver Carcinoma (HepG2) Cells

**DOI:** 10.3390/ijerph192114272

**Published:** 2022-11-01

**Authors:** Guofa Ren, Kangming Wu, Jing An, Yu Shang, Kewen Zheng, Zhiqiang Yu

**Affiliations:** 1Institute of Environmental Pollution and Health, School of Environmental and Chemical Engineering, Shanghai University, Shanghai 200444, China; 2State Key Laboratory of Organic Geochemistry, Guangdong Key Laboratory of Environment Protection and Resource Utilization, Guangzhou Institute of Geochemistry, Chinese Academy of Sciences, Guangzhou 510640, China

**Keywords:** octachlorostyrene, HepG2 cells, oxidative stress, cell apoptosis

## Abstract

Octachlorostyrene (OCS) is a ubiquitous persistent organic pollutant; however, information regarding the toxicological effects of OCS remains limited. In this study, we studied the toxicity mechanisms of OCS using human liver carcinoma (HepG2) cells. The results showed that OCS reduced cell viability in a time- and dose-dependent manner. Compared with that in the control, the level of reactive oxygen species (ROS) was significantly increased in all treated HepG2 cells. We also found that (1) OCS induced damage in the HepG2 cells via the apoptotic signaling pathway, (2) OCS increased intracellular free Ca^2+^ concentration (>180%), and (3) following exposure to 80 μM OCS, there was an increase in mitochondrial transmembrane potential (MMP, ~174%), as well as a decrease in ATP levels (<78%). In conclusion, OCS is cytotoxic and can induce apoptosis, in which ROS and mitochondrial dysfunction play important roles; however, the observed increase in MMP appears to indicate that HepG2 is resistant to the toxicity induced by OCS.

## 1. Introduction

Octachlorostyrene (OCS, Chemical Abstracts Service registry number 29082-74-4) has the empirical molecular formula C_8_Cl_8_ and a molecular weight of 379.71 g/mol. As can be seen from the structural formula shown in Figure 1, OCS is obtained when all the hydrogen atoms of styrene are substituted with chlorine atoms.

OCS has never been commercially produced anywhere in the world, but it is formed as a byproduct in many chlorinated industrial processes, such as PVC recycling activities, aluminum refining operations, and metal-chlorinated solvent degreasing operations. OCS is released through effluent discharges from high temperature chlorine industry facilities and subsequently spreads through the water cycle. According to a report by the United States Environmental Protection Agency (USEPA), OCS has the typical characteristics of a persistent organic pollutant: high lipophilicity, persistency in the environment, and potential for accumulation in organisms. This belief is corroborated by a series of reports showing that OCS has been detected in fish, seafood, birds, mammals, and other organisms [1,2,3,4]. OCS has been shown to bioaccumulate at levels comparable to those of flame retardants in wildlife of the same species collected at the same location [2]. Temporal trends show that the concentration of OCS in some wildlife species has increased in recent years [3]. In addition, studies have demonstrated that OCS can bioaccumulate in the human body (such as in breast milk and blood) [5,6,7], which provides a meaningful motivation for the health assessment of OCS.

Although OCS is classified by the USEPA as a persistent and bioaccumulative toxic chemical, information about its toxic impacts and mode of action is still very limited. Park et al. reported that OCS induces apoptotic cell death in human Chang liver cells via ROS increase and cytosolic caspase-3 activation [8]. Evidence from animal studies suggests that high-dose OCS in rats may induce an increase in liver weight and heme biosynthesis [9]. In addition, levels of microsomal protein, cytochrome P450, cytochrome P450 reductase, ethylmorphine n-demethylase, 4-nitroanisole o-demethylase, and acetanilide 4-hydroxylase increased in the rats after OCS treatment. Several researchers have studied the mechanisms underlying its toxicity. OCS can directly activate the aryl hydrocarbon receptor (AhR) and constitutive androstane receptor (CAR), increasing the transcription of *CYP1A* and *CYP2B* [10]. Kamata et al. used a yeast cell reporter gene assay to determine the agonistic effect of different exogenous compounds on CAR and found that OCS was one of the ten most potent compounds [11]. A whole genome microarray of *Caenorhabditis elegans* has been used to observe and assess chronic toxicity caused by exposure to OCS. OCS affected the reproductive rate of *C. elegans*, and the authors identified 1294 genes that were differentially expressed across generations [12].

Presently, little is known about the detrimental effects of OCS at the molecular and cellular levels, and studies on the effects of OCS in different cell types are required. In this study, we investigated the cellular and molecular mechanisms of OCS-induced cytotoxicity in HepG2 cells. Toxicity was assessed using a range of in vitro assays to study multiple mechanisms that may contribute to the toxicity caused by OCS exposure. The assays included both MTT and CCK8, as well as the measurement of the lactate dehydrogenase (LDH) leakage rate, ROS production, and the activity of enzymes associated with oxidative stress (superoxide dismutase [SOD], glutathione assay [GSH], malondialdehyde [MDA], and catalase [CAT] quantification). In addition, intracellular free calcium ion (Ca^2+^) concentrations, adenosine triphosphate (ATP) levels, and mitochondrial transmembrane potential (MMP) were determined to further understand the mechanisms of OCS-induced toxicity in HepG2 cells. The data may provide an important reference for subsequent studies on environmental toxicity, as well as an ecological risk assessment of OCS.

## 2. Materials and Methods

### 2.1. Chemicals and Consumables

Octachlorostyrene (99.4% purity) was purchased from Dr. Ehrenstorfer GmbH (Ausburg, Germany). Dulbecco’s modified Eagle’s medium (DMEM) was obtained from Biological Industries (BI, Cromwell, CT, USA), as was fetal bovine serum (FBS) and penicillin/streptomycin. DMSO, 3-(4,5-Dimethylthiazol-2-yl)-2,5-diphenyltetrazolium bromide (MTT), an Annexin-FITC/PI cell apoptosis detection kit, an ATP detection kit, a Catalase Assay kit, a Total Superoxide Dismutase Assay Kit with WST-8, a Cell Counting Kit-8, a Cellular Glutathione Peroxidase Assay Kit with NADPH, Fluo-4 AM, Rhodamine 123, a LDH Cytotoxicity Assay Kit, and a Lipid Peroxidation MDA Assay Kit were obtained from the Beyotime Institute of Biotechnology (Beyotime Biotechnology, Shanghai, China). The 2′, 7′-dichlorofluorescein-diacetate (DCFH-DA) probe was obtained from Sigma-Aldrich (St. Louis, MO, USA). Culture plastics were purchased from Corning Costar Corporation (Cambridge, MA, USA).

### 2.2. Cell Culture

The human hepatocellular carcinoma cell line HepG2, cultured in DMEM (containing 1% streptomycin/penicillin and 10% fetal bovine serum) at 37 °C in a humidified atmosphere containing 5% carbon dioxide, was provided by American Type Culture Collection (ATCC). The OCS was dissolved in DMSO, and the solvent control, of which the final concentration of DMSO was 0.1%, was established.

### 2.3. Cell Viability Assay

Cell viability was evaluated by MTT and CCK8 assay. The cell lines HepG2 (4 × 10^3^) were seeded in 96-well culture plates for 16 h at 37 °C, under a humidified atmosphere in a CO_2_ incubator. The cells were treated with a series of OCS concentrations (10 μM, 20 μM, 40 μM, 80 μM) and 0.1% DMSO (as the control) to assess cytotoxicity, and 0.1% DMSO was used as a solvent control. After exposure with OCS for 12, 24, 36, or 48 h, 100 mL of MTT reagent (5 mg/mL) was added to each well, the plate was incubated in the CO_2_ incubator for 4 h, the supernatant was discarded, and 150 mL DMSO was added to each well. Then, we gently shook the 96-well plate for 3 min in the dark on a shaker platform to dissolve the formazan crystals. The formazan crystals were recorded by absorbance at 490 nm using a microplate reader (Spark^®^ 20M Multimode Microplate Reader, Tecan, Männedorf, Switzerland).

Additionally, after OCS 24 h treatment, CCK8 for was also detected using WST-8, which produces corresponding formazan dye. Then, 10 μL of Cell Counting Kit-8 reagent was added to each well on the 96-well microplate. The reaction occurred for 1–4 h in a CO_2_ incubator. Absorbance was recorded at 450 nm using a microplate reader (Spark^®^ 20M Multimode Microplate Reader, Tecan, Switzerland).

### 2.4. Membrane Integrity Assay

The LDH release assay, a cell death/cytotoxicity assay, plays an important role in assessing the level of plasma membrane disruption. In short, cells with a density of 4 × 10^3^ were collected and cultured in each well of a 96-well plate for 16 h at 37 °C, under a humidified atmosphere in a CO_2_ incubator, and then treated with a series of OCS concentrations (10 μM, 20 μM, 40 μM, or 80 μM) and 0.1% DMSO (as a control) for 24 h. Next, LDH reagent (1:100) was added to each well and incubated in the incubator for 1 h. After incubation, the cells were centrifuge at 400× *g* for 5 min, and 120 μL of supernatant from each well was transferred to a new 96-well plate, and 60 μL of LDH detection reagent was added to each well. After incubation for 30 min in the dark, absorbance was recorded at 490 nm using a microplate reader (Spark^®^ 20M Multimode Microplate Reader, Tecan, Switzerland).

### 2.5. Measurement of ROS and Several Other Oxidative Stress Parameters

We added the cell suspension with a density of 1.5 × 10^5^ cells to each well in a 6-well plate. After adhering and reaching 70–80% confluence, the cells were exposed to a range of concentrations of OCS (10 μM, 20 μM, 40 μM, 80 μM) and 0.1% DMSO (as a control) for 4 h, 12 h, or 24 h. Once finished, the culture was removed and the HepG2 cells were further incubated with 2′, 7′-dichlorofluorescein-diacetate (DCHF-DA) for 30 min in the dark in an incubator at 37 °C. The diluted DCFH-DA solution was removed, and PBS buffer was used to wash the HepG2 cells three times. Cell imager (ZOE™ Fluorescent Cell Imager, BIO-RAD, Hercules, CA, USA) was used to view and photograph the fluorescent images. Image-Pro Plus (Medical Cybernetics, Version 6.0, Rockville, MD, USA) was used to analyze and quantify the fluorescence intensities.

Several oxidative stress parameters, such as superoxide dismutase (SOD), glutathione assay (GSH), malondialdehyde (MDA), and catalase (CAT) were evaluated in the HepG2 cells at different concentrations of OCS, between 0 and 80 μM, at 20 μM intervals for 24 h. A series of assay kits (Beyotime Institute of Biotechnology, Shanghai, China) were used to detect the activities of several oxidative stress parameters. After exposure, the culture medium was aspirated and discarded, and the cells were washed gently with PBS, pre-cooled at 4 °C. The sample homogenate provided in the kit was added at a ratio of 100–200 μL per 1 million cells. After homogenization or lysis, the cells were centrifuged at 10,000–12,000× *g* for 10 min and the supernatant was collected for subsequent assays. sample wells and blank control wells were set up using a 96-well plate. the appropriate reagent solution was added to each well, according to the manufacturer’s directions, and mixed well. The absorbances recorded for the SOD, GSH, MDA, and CAT assays were 450 nm, 340 nm, 532 nm, and 520 nm, respectively, using a microplate reader (Spark^®^ 20M Multimode Microplate Reader, Tecan, Switzerland). All assays were replicated three times. The final calculation of enzyme activity in the samples was carried out.

### 2.6. MMP and ATP Quantification

Cells were seeded in 6 wells following a density of 1.5 × 10^5^ cells per well for both experiments. After exposure to OCS (10 μM, 20 μM, 40 μM and 80 μM), changes in MMP and ATP quantification were detected with fluorescent cationic dye Rhodamine 123 (Beyotime Biotech) and an ATP Assay Kit (Beyotime Biotech), respectively, complying with the manufacturer’s directions. A cell imager (ZOE™ Fluorescent Cell Imager, BIO-RAD, USA) was used to view and photograph the fluorescence images of MMP. Image-Pro Plus (Medical Cybernetics, Version 6.0) was used to analyze and quantify the fluorescence intensities. Additionally, the ATP signal was detected using a microplate reader (Spark^®^ 20M Multimode Microplate Reader, Tecan, Switzerland).

### 2.7. Intracellular Ca^2+^ Measurements

After being treated with OCS, the HepG2 cells were co-incubated at room temperature with Fluo-4 AM for 25 min in the dark. Once finished incubation, using PBS buffer to wash HepG2 cells three times and then suspended in PBS (1 mL). Fluorescence images of intracellular Ca^2+^ were viewed and photographed under a cell imager (ZOE™ Fluorescent Cell Imager, BIO-RAD, USA). Image-Pro Plus (Medical Cybernetics, Version 6.0) was used to analyze and quantify the fluorescence intensities.

### 2.8. Apoptosis and AnnexinV-FITC/PI Assay

HepG2 cells were seeded in 6-well plates at a density of 1.5 × 10^5^ per well. We used EDTA-free trypsin to digest HepG2 cells, with a cell density of 70–80%, in 6-well plates, assaying them for apoptosis using the Annexin V-FITC/PI kit (Beyotime Biotech). The collected cells were washed twice with PBS, and then centrifuged at 4 °C for 5 min in a refrigerated high-speed centrifuge. Then, 100 mL 1× binding buffer was used to resuspend HepG2, and 5 mL Annexin V-FITC and 5 mL PI Staining Solution were both incubated in the dark for 10 min at room temperature. Next, 400 mL 1× binding buffer was added, and a Flow Cytometer (CytoFLEXs, Beckman Coulter, Indianapolis, IN, USA) was used to analyze the apoptosis.

### 2.9. Statistical Analyses

GraphPad Prism (GraphPad Prism 6.0, San Diego, CA, USA) was used for statistical analysis. All experiments were carried out in three independent replications, and the data obtained are shown by mean ± standard deviation. One-way analysis of variance (ANOVA) was used to perform the statistical analysis of the data, followed by a *t*-test. The *p*-value ≤ 0.05 indicates statistical significance.

## 3. Results and Discussions

### 3.1. Cell Viability and Cytotoxicity

First, we used MTT and CCK8 assays to assess cytotoxicity related to OCS exposure in HepG2 cells. In the MTT assay, toxicity in HepG2 cells was assessed at multiple time points (12, 24, 36, and 48 h) over a range of OCS concentrations. The exposure groups with the lowest concentration of OCS (10 μM) showed slight toxicity compared to the control group, except for the 12 h exposure group. Clear time-dependent effects were observed in the 20, 40, and 80 μM exposure groups (Figure 1A). Furthermore, OCS reduced the viability of HepG2 cells in a dose-dependent manner. The cytotoxicity of OCS on HepG2 cells at 24 h was also demonstrated by the results of the CCK8 assay (Figure 1B). A previous study demonstrated that OCS exhibited time- and dose-dependent cytotoxicity to human Chang liver cells at similar concentrations; however, the lower concentration treatment group (6.25 μM) did not have an evident effect on cell viability after 24 h of exposure [8]. To further quantify the cytotoxicity of OCS in HepG2 cells, LDH leakage was measured. LDH is a critical characteristic of cells undergoing apoptosis and other forms of cellular damage. When the plasma membrane is damaged by external stimuli, the LDH enzyme is rapidly released [13]. In our study, LDH leakage was significantly higher in the 20, 40, and 80 μM OCS-treated groups (Figure 1C), suggesting cell membrane damage at these three concentrations. Interestingly, LDH activity was lower at 80 μM compared to that at 40 μM, which might be due to the increase in cell death. Based on these findings, we used four concentrations of OCS (10, 20, 40, and 80 μM) in the subsequent experiments, as 80 μM OCS was significantly cytotoxic to HepG2 cells, but did not cause significant cell death.

### 3.2. ROS Level and Antioxidant Parameters in HepG2 Cells

Oxidative stress plays an essential role in many human diseases. The overproduction of ROS damages cells and has been shown to be involved in neurodegenerative diseases [14], cancer [15], heart disease [16], and osteoporosis [17]. Moreover, oxidative stress plays key roles in fundamental biological processes; increased ROS production can induce DNA damage and affect the DNA damage response [18]. To determine whether OCS induces ROS overproduction, leading to oxidative stress events and hence, cytotoxicity, we measured total intracellular ROS generation using DCFH-DA. As shown in Figure 2, ROS levels were measured after 4, 12, and 24 h of OCS treatment, and both time- and dose-dependent effects were observed. Compared with that in the control group, the fluorescence intensity was significantly higher in all OCS-treated groups (Figure 2B). Oxidative stress results from the toxic effects of various environmental pollutants, including organophosphate flame retardants. For example, a previous study found that TPHP caused high ROS levels after exposure for 96 h, resulting in cell membrane damage in marine microalga *Chaetoceros meülleri* [19]. In this study, we demonstrated that OCS significantly promoted ROS overproduction in HepG2 cells and consequently induced an oxidative damage response (Figure 2).The body houses a complex antioxidant defense grid to maintain a normal oxidative balance, which relies on endogenous enzymatic and non-enzymatic antioxidants [20]. These molecules collectively play a central role in removing active oxygen. In this study, antioxidant parameters, such as GSH and CAT, were measured in HepG2 cells after OCS treatment. GSH levels decreased significantly between OCS treatment at 10 μM and the highest concentration of 80 μM (Figure 3A). SOD activity was not significantly altered compared with that in the control group (Figure 3B). MDA production in HepG2 cells was markedly higher after treatment with 80 μM OCS, showing a 210.38% increase compared with that in the control (Figure 3C). Following treatment with 20, 40, and 80 μM OCS, CAT activity decreased by 7.09%, 25.75%, and 37.50%, respectively (Figure 3D). Reduced antioxidant parameters may reflect an imbalance in, or the overwhelming of, the ROS antioxidant system. GSH deficiency, or a decreased GSH/glutathione disulfide (GSSG) ratio, is primarily manifested by the activation of the antioxidant system and increased sensitivity to oxidative stress [21]. Previous studies on heavy metals in environmental samples have found that GSH aids in the detoxification of these toxicants within the cells, whereas antioxidant enzymes play a central role in removing active oxygen, resulting in a decrease in GSH or the GSH/GSSH ratio [22,23]. We observed unchanged SOD activity, which may indicate insensitivity of SOD to OCS. If the initial overproduction of superoxide due to SOD1 insensitivity cannot be eliminated effectively in time, GSH compensates for the effects of SOD1 inactivity or reduced activity, thus protecting cells from oxidative stress and death [24]. We also found a decrease in CAT; this may be due to the fact that CAT and GSH serve as the primary lines of defense against oxidative stress, whereas SOD provides an insensitivity to OSC-induced oxidative stress. Overall, these results demonstrate that OCS imparted a high toxicity to HepG2 cells as a consequence of the overproduction of ROS.

Oxidative stress plays an essential role in many human diseases. The overproduction of ROS damages cells and has been shown to be involved in neurodegenerative diseases [14], cancer [15], heart disease [16], and osteoporosis [17]. Moreover, oxidative stress plays key roles in fundamental biological processes; increased ROS production can induce DNA damage and affect the DNA damage response [18]. To determine whether OCS induces ROS overproduction, leading to oxidative stress events and hence, cytotoxicity, we measured total intracellular ROS generation using DCFH-DA. As shown in Figure 2, ROS levels were measured after 4, 12, and 24 h of OCS treatment, and both time- and dose-dependent effects were observed. Compared with that in the control group, the fluorescence intensity was significantly higher in all OCS-treated groups (Figure 2B). Oxidative stress results from the toxic effects of various environmental pollutants, including organophosphate flame retardants. For example, a previous study found that TPHP caused high ROS levels after exposure for 96 h, resulting in cell membrane damage in marine microalga *Chaetoceros meülleri* [19]. In this study, we demonstrated that OCS significantly promoted ROS overproduction in HepG2 cells and consequently induced an oxidative damage response (Figure 2).The body houses a complex antioxidant defense grid to maintain a normal oxidative balance, which relies on endogenous enzymatic and non-enzymatic antioxidants [20]. These molecules collectively play a central role in removing active oxygen. In this study, antioxidant parameters, such as GSH and CAT, were measured in HepG2 cells after OCS treatment. GSH levels decreased significantly between OCS treatment at 10μM and the highest concentration of 80 μM (Figure 3A). SOD activity was not significantly altered compared with that in the control group (Figure 3B). MDA production in HepG2 cells was markedly higher after treatment with 80 μM OCS, showing a 210.38% increase compared with that in the control (Figure 3C). Following treatment with 20, 40, and 80 μM OCS, CAT activity decreased by 7.09%, 25.75%, and 37.50%, respectively (Figure 3D). Reduced antioxidant parameters may reflect an imbalance in, or the overwhelming of, the ROS antioxidant system. GSH deficiency, or a decreased GSH/glutathione disulfide (GSSG) ratio, is primarily manifested by the activation of the antioxidant system and increased sensitivity to oxidative stress [21]. Previous studies on heavy metals in environmental samples have found that GSH aids in the detoxification of these toxicants within the cells, whereas antioxidant enzymes play a central role in removing active oxygen, resulting in a decrease in GSH or the GSH/GSSH ratio [22,23]. We observed unchanged SOD activity, which may indicate insensitivity of SOD to OCS. If the initial overproduction of superoxide due to SOD1 insensitivity cannot be eliminated effectively in time, GSH compensates for the effects of SOD1 inactivity or reduced activity, thus protecting cells from oxidative stress and death [24]. We also found a decrease in CAT; this may be due to the fact that CAT and GSH serve as the primary lines of defense against oxidative stress, whereas SOD provides an insensitivity to OSC-induced oxidative stress. Overall, these results demonstrate that OCS imparted a high toxicity to HepG2 cells as a consequence of the overproduction of ROS.

### 3.3. Effect on MMP and ATP Level

Disruption of mitochondrial integrity is well known as one of the key early events in the apoptotic cascade. Therefore, we investigated whether mitochondrial function is altered in HepG2 cells after OCS treatment in vitro. We then determined whether apoptosis is altered in HepG2 cells by mitochondrial stress by measuring the MMP and ATP levels after treatment with OCS. It was found that the MMP increased after 24 h of treatment with OCS (Figure 4A), and the intracellular ATP level decreased only at the highest concentration of 80 μM (Figure 4B). The MMP increased by 174.03% in the highest OCS concentration-treated group compared with that in the control group. Mitochondrial dysfunction is closely associated with many human diseases, for example, cancer and neurodegenerative diseases [25,26]. Numerous studies focused on ROS-dependent pathways in the mitochondria created in response to exogenous compounds may provide new insights into environmental pollutant-induced toxicity [27,28,29]. Recently, Sun et al. found that mitochondrial fusion/fission dynamics in PK15 cells are disrupted by BDE-47, which in turn induces mitochondrial abnormalities and triggers oxidative stress, leading to cell dysfunction [30]. In addition, researchers have found that exposure to organohalogens results in ROS overproduction via a molecular initiating event that triggers the key events causing the adverse outcomes that are seen at the onset of mitochondrial disease [31]. To research whether mitochondrial dysfunction is involved in OCS-induced cytotoxicity, the levels of MMP were examined. As shown in Figure 4, OCS significantly increased MMP, which may demonstrate a mechanism by which HepG2 cells resist OCS-induced cell death. In addition, ATP levels, compared before and after OCS exposure, revealed that ATP levels decreased, but were not completely depleted, at the highest dose of OCS (80 μM) for 24 h. Previous studies have found that in the early phase of glutamate toxicity, geranylgeranyl acetone treatment increases MMP. In this manner, mitochondrial calcium buffering capacity is enhanced as a defense against reactive oxygen species-induced calcium overload [32]. Ma et al. reported that low oxygen tension may provide an environment conducive to viability by the upregulation of LIFR/VEGF and an increase in MMP, which could enhance implantation potential and apoptosis resistance in mouse blastocysts [33]. Conversely, this also means that cells with enhanced MMP drive tumor formation more efficiently, and therefore, MMP may be a marker for cancer-initiating cells [34]. These results suggest that OCS may cause mitochondrial dysfunction and that ROS overproduction may have contributed to changes in the normal state of MMP, causing it to be upregulated to resist oxidative stress induced by OCS.

### 3.4. Intercellular Ca^2+^ and Apoptosis

Calcium ions function as secondary messengers and regulate numerous intracellular events, including muscle contraction, exocytosis, metabolism, neurotransmitter release, and gene transcription [35]. We assessed intercellular Ca^2+^ levels in HepG2 cells before and after OCS treatment. As shown in Figure 5, all OCS-treated groups showed increased fluorescence intensity, which indicated high levels of intercellular Ca^2+^, compared with the control group. However, no dose-dependent effect was observed, with the different treatment groups (10, 20, 40, and 80 μM) showing increased intercellular Ca^2+^ levels of 229.60%, 186.69%, 234.11%, and 189.37%, respectively. These results reveal that the changes in intercellular Ca^2+^ concentration caused by OCS exposure accompany the onset of apoptosis. Intracellular Ca^2+^ signaling, cell survival, and cell death are closely linked through the regulation of apoptosis, mitochondrial bioenergetics, and autophagy. Intracellular Ca^2+^ signaling and Ca^2+^ microdomains can activate apoptosis and autophagy in response to cellular damage or stress caused by exogenous compounds. Moreover, the Ca^2+^ microdomain structure can control critical cell survival/death decisions [36]. Subcellular regions with high Ca^2+^ concentrations form Ca^2+^ microdomains that develop rapidly near open Ca^2+^ channels in the plasma membrane or internal stores, and in turn, generate localized regions of high Ca^2+^ concentration [37]. Numerous studies have shown that the toxicity of environmental pollutants is usually accompanied by an increase in intracellular Ca^2+^, leading to an imbalance in intracellular Ca^2+^ homeostasis and possibly inducing the formation of Ca^2+^ microdomains, which may affect important functional roles in life activities [38]. For example, impairment of normal calcium regulation by combustion-derived particles (CDP) may be a key cellular event; CDP exposure contributes to the development or aggravation of cardiovascular disease by affecting intracellular calcium regulation events [39]. In addition, a rapid AhR-dependent Ca^2+^ increase in HMEC-1 endothelial cells was induced by lipophilic constituents of diesel exhaust particles extracted by n-hexane and DCM, possibly involving both ROCE and SOCE-mediated mechanisms [40]. Sarco/endoplasmic reticulum Ca^2+^ dynamics, triggering Ca^2+^ efflux from microsomal vesicles, were altered by organohalogens naturally biosynthesized in marine environments and produced as disinfection byproducts [41]. Furthermore, methylmercury-induced cytotoxicity is associated with mitochondrial dysfunction via Ca^2+^ overload. Mitochondrial disturbances caused by the imbalance of intracellular Ca^2+^ homeostasis may involve numerous molecular events, such as mitochondrial permeability transition pore opening, ultimately pointing to cell death. Overall, we found that the intracellular free Ca^2+^ concentration in the HepG2 cells was significantly increased after 24 h of OCS exposure, suggesting that the intracellular Ca^2+^ signaling pathway is involved in OCS-induced cytotoxicity.

### 3.5. OCS Induces Apoptosis in HepG2 Cells

Apoptosis is a form of programmed cell death that ensures a homeostatic balance between the rate of cell formation and cell death through the orderly and efficient removal of damaged cells. In this study, annexin FITC/PI staining assays were performed to confirm whether OCS triggered apoptosis. As shown in Figure 6A, the HepG2 cell apoptosis rate significantly increased after treatment with 40 and 80 μM OCS compared with that of the control. Following exposure to 80 μM OCS for 24 h, the number of combined late apoptotic and early apoptotic cells increased from 6.39% to 14.14%, respectively, compared with that of the control. Imbalance in apoptotic homeostasis is closely associated with numerous diseases, such as cancer, autoimmune diseases, neurological disorders, and cardiovascular disorders [42]. Numerous studies have found that chemical compounds, including drugs and environmental pollutants, induce cell apoptosis by promoting ROS production via the ROS-mediated mitochondrial pathway, caspase activation, DNA damage, the NF-κB signaling pathway, and the JNK pathway [43,44,45]. We observed the overproduction of ROS and suspect that ROS may mediate the development of adverse outcome pathways that may ultimately lead to apoptosis. Our results revealed a positive correlation between the level of apoptosis and the OCS concentration, with significant OCS-induced apoptosis being observed at doses of 40 and 80 μM (Figure 6B). Although oxidative stress stimulated by exogenous compounds can induce apoptosis, directly or indirectly, through multiple pathways, we speculate that OCS-induced apoptosis may be caused directly by the overproduction of ROS in the intracellular mitochondria, leading to the onset of oxidative stress events. In conclusion, our data provide evidence that apoptosis, as well as oxidative stress, are possibly involved in OCS-induced HepG2 cytotoxicity.

## 4. Conclusions

Our results demonstrated that OCS induced cytotoxicity in HepG2 cells and that OCS stimulated ROS overproduction, causing oxidative stress events, which in turn stimulated the antioxidant defense system in HepG2 cells. However, the mitochondrial transmembrane potential also significantly increased after OCS treatment, which may be due to the activation of a defense mechanism in HepG2 cells in response to the toxicity of OCS exposure. Exposure to 80 μM OCS induced significant apoptosis, which was accompanied by an increase in the intracellular Ca^2+^ concentration. We hypothesize that the underlying mechanism of OCS-induced toxicity in HepG2 cells is related to the generation of ROS and that the excessive production of ROS may have led to mitochondrial dysfunction, initiating the apoptotic cascade process in HepG2 cells, accompanied by an increase in the intracellular Ca^2+^ concentration. In the future, we need to analyze the pathways involved in OCS-induced apoptosis in HepG2 cells; for example, we could use gene sequencing and gene silencing technologies to identify both the initiation mechanism of the OCS-induced apoptotic pathway and the key genes and proteins involved.

## Figures and Tables

**Figure 1 ijerph-19-14272-f001:**
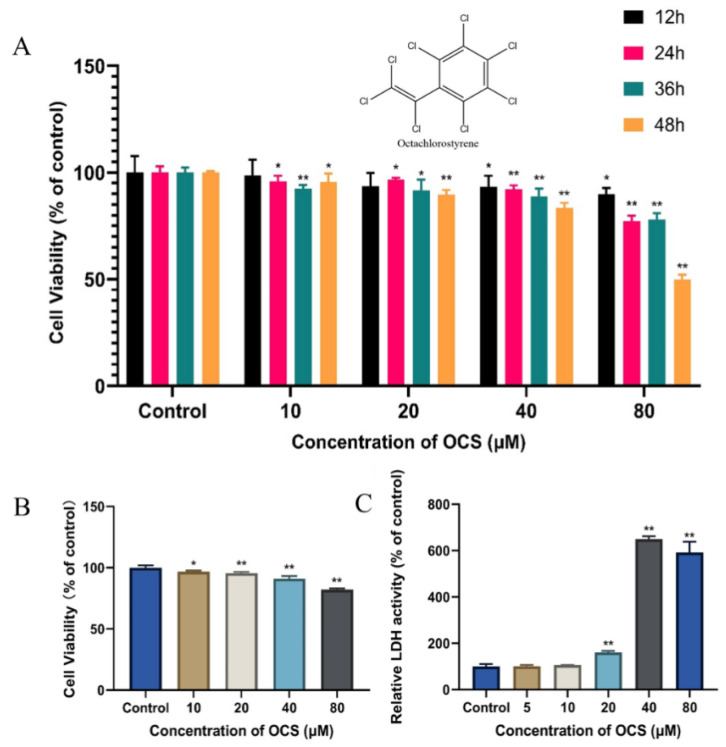
The cell viability of HepG2 cells after treatment with OCS for 12–48 h based upon the MTT assay (**A**) and the CCK8 assay (**B**). The LDH release (**C**) of HepG2 cells was performed after treatment with OCS for 24 h. Each value represents the mean ± S.D. of three independent experiments (*n* = 3). * (*p* < 0.05) or ** (*p* < 0.001) denote a significant difference relative to the control group.

**Figure 2 ijerph-19-14272-f002:**
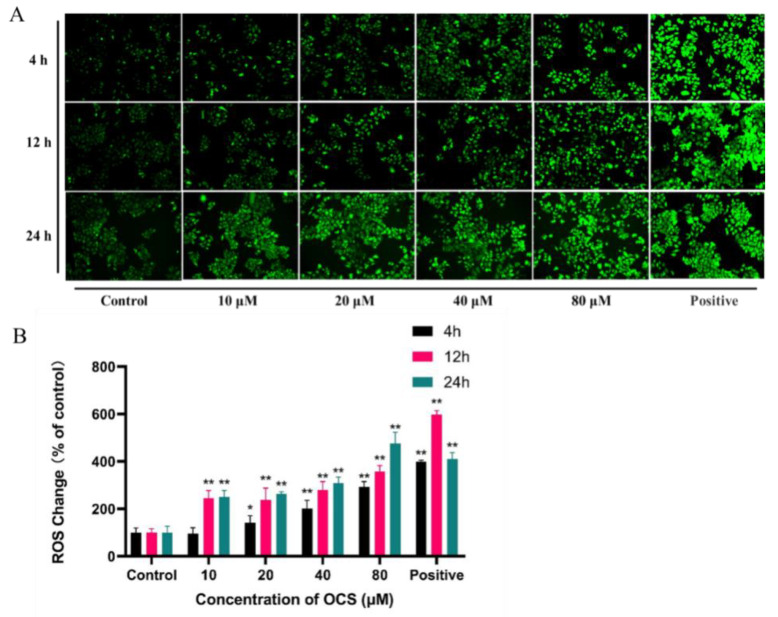
The image (**A**) and change (**B**) of intracellular ROS generation was detected by DCHF-DA after treatment with OCS for 4 h, 12 h, and 24 h. Each value represents the mean ± S.D. of three independent experiments (*n* = 3) (* *p* < 0.05 vs. control, ** *p* < 0.001 vs. control).

**Figure 3 ijerph-19-14272-f003:**
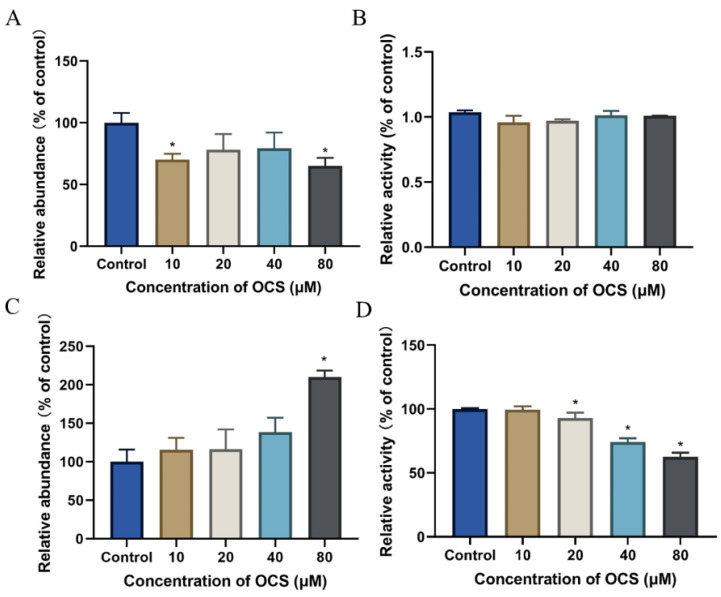
OCS-induced oxidative stress in HepG2 cells after exposed for 24 h. (**A**) Glutathione depletion; (**B**) Superoxide dismutase; (**C**) Lipid peroxidation based on MDA; (**D**) Catalase activity; Values are mean ± SD of three independent experiments (*n* = 3). (* *p* < 0.05 vs. control).

**Figure 4 ijerph-19-14272-f004:**
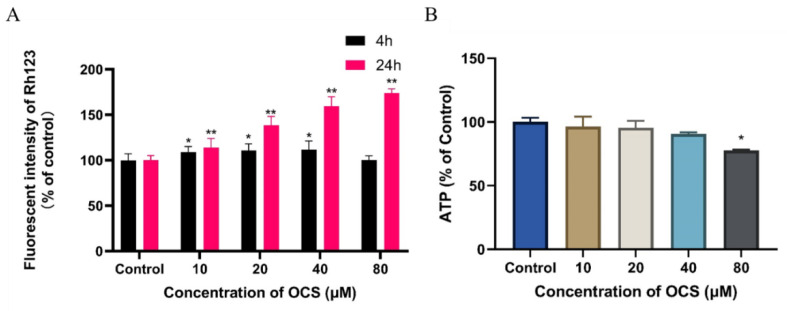
Indicators of mitochondrial dysfunction in HepG2 cells following OCS exposure for 24 h; (**A**) increase in MMP after treatment with OCS in HepG2 cells; (**B**) OCS decreased the generation of ATP in HepG2 cells. Values are mean ± SD of three independent experiments (*n* = 3) (* *p* < 0.05, ** *p* < 0.001 vs. control).

**Figure 5 ijerph-19-14272-f005:**
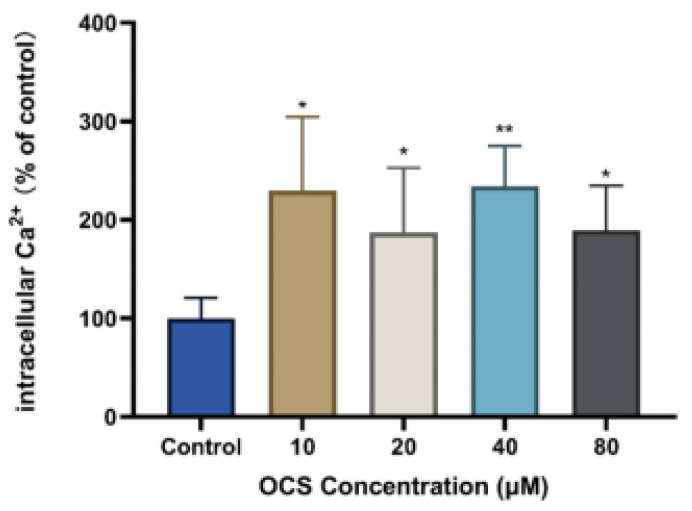
The concentration of intracellular Ca^2+^ after treatment with OCS for 24 h in HepG2 cells. Values are mean ± SD of three independent experiments (*n* = 3) (* *p* < 0.05, ** *p* < 0.001 vs. control).

**Figure 6 ijerph-19-14272-f006:**
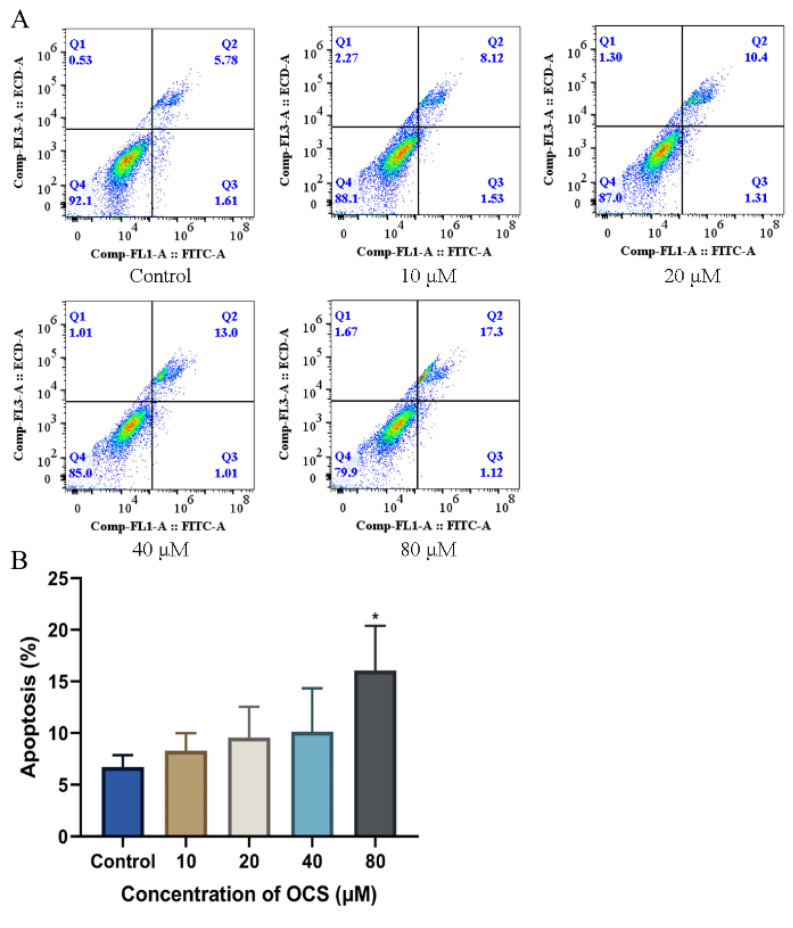
Induction of apoptosis by OCS in HepG2 cells after 24 h exposure. (**A**) Apoptosis in HepG2 cells after treatment with OCS for 24 h was assessed by Annexin-FITC/PI staining. (**B**) Percent of apoptotic HepG2 cells induced by OCS following 24 h treatment. Values are mean ± SD of three independent experiments (*n* = 3) (* *p* < 0.05 vs. control).

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
