# Peer review of "Toxicity Assessment of Octachlorostyrene in Human Liver Carcinoma (HepG2) Cells"

_ijerph, 2022, doi:10.3390/ijerph192114272_

Round 1

Reviewer 1 Report

The author's manuscript on ‘Toxicity assessment of octachlorostyrene in human liver carcinoma (HepG2) cell’ has sufficient evidence to show that OCS induces oxidative stress, mito dysfunction, and apoptosis in HepG2 cells at higher concentrations. However, the manuscript needs minor corrections before accepting to the International Journal of Environmental Research and Public Health.

The review comments are as follows:

1.     Authors need to include assay-specific controls for the ROS assay. For example, treat 80µM OCS samples with NAC or membrane permeable catalase and measure the DCFHDA fluorescence.

2.     Authors need to treat HepG2 cells with either CCCP or FCCP, which affects mitochondrial membrane potential as an assay-specific control for MMP assay.

3.      Authors need to treat HepG2 cells with tapsigargin, as an assay-specific control for intracellular calcium measurements.

4.     The X- and Y-axis labels are not visble in Figur 6. Dot plot images. Please revise them. 

Author Response

Reviewer: 1

The specific comments are as follows:

  1. Authors need to include assay-specific controls for the ROS assay. For example, treat 80µM OCS samples with NAC or membrane permeable catalase and measure the DCFHDA fluorescence.
  2. Authors need to treat HepG2 cells with either CCCP or FCCP, which affects mitochondrial membrane potential as an assay-specific control for MMP assay.
  3. Authors need to treat HepG2 cells with tapsigargin, as an assay-specific control for intracellular calcium measurements.

Response: Thank you for the suggestion. The above three questions are all about the addition of assay-specific control samples, which we believe can be explained together. Because the regulatory mechanisms of acellular messengers are very complex. Take the first question posed by the reviewer as an example, numerous other cellular sources of ROS in the recent literature, among which mitochondria and NADPH oxidase are the two major sources. Different sources of ROS have different specific antioxidant agents. The main purpose of this study was to explore the dose and time effects of exogenous pollutants on cytotoxicity, without in-depth analysis of its biological mechanism, so no specific control samples were selected. In fact, we used the tests commonly used in most of the literature on the cytotoxicity of exogenous pollutants.

  1. The X- and Y-axis labels are not visble in Figur 6. Dot plot images. Please revise them. 

Response: Sorry for our carelessness. For the sake of clarity, we redrew Figure 6 in our revised manuscript, please check.

Reviewer 2 Report

The manuscript  entitled “Toxicity assessment of octachlorostyrene in human liver carcinoma (HepG2) cells” present the results of toxicity mechanisms of OCS using human liver carcinoma (HepG2) cells.

Authors should correct manuscript according to the suggestion.

Major issues:

Materials and Methods:

2.2.  Please explain why HepG2 carcinoma cell line was used for estimation of OCS toxicity? Authors used any other line as a control? cell lines, especially normal and carcinoma cells were different metabolism, especially in case of chemicals (include xenobiotics). It is resulted by higher cytochrome genes activity in carcinoma cells and faster xenobiotics metabolism.

2.5. please give information how SOD, CAT, GSH, MDA were measured, in substrate concentration, negative and positive control and absorbance

2.3; 2.4; 2.6; 2.7 and 2.8 what were the controls in experiments? please add information (on Figures controls were placed)

Author Response

The specific comments are as follows:

Materials and Methods:

2.2.  Please explain why HepG2 carcinoma cell line was used for estimation of OCS toxicity? Authors used any other line as a control? cell lines, especially normal and carcinoma cells were different metabolism, especially in case of chemicals (include xenobiotics). It is resulted by higher cytochrome genes activity in carcinoma cells and faster xenobiotics metabolism.

Response:  We appreciate this professional suggestion. Explanation:HepG2 cells are derived from human hepatoblastoma, and the biometabolic invertase contained in HepG2 cells has homology with human liver normal cells, which has a high degree of differentiation, and retains a relatively complete biometabolic invertase. Therefore, it is not necessary to add an exogenous activation system when using this cell for cytotoxicity experiment. In contrast, human primary hepatocytes can only undergo a limited number of divisions after separation, and their intrinsic active enzymes quickly lose their activity. HepG2, as a well-differentiated cell line, has stable intrinsic metabolic enzyme activity and will not lose its activity with the increase of passage times.

2.5. please give information how SOD, CAT, GSH, MDA were measured, in substrate concentration, negative and positive control and absorbance

Response: We highly appreciate the reviewer for this constructive comment. To be more clear and in accordance with the reviewer concerns, we have added a brief description of the measurement as below:Several oxidative stress parameters, such as superoxide dismutase (SOD), glutathione assay (GSH), malondialdehyde (MDA) and catalase (CAT) in HepG2 cells were evaluated at different concentrations of OCS, between 0 and 80 μM at 20 μM intervals for 24 h. A series of assay kits (Beyotime Institute of Biotechnology, China) were applied to detect the activities of several oxidative stress parameters.  After exposure, aspirate and discard culture medium and wash the cells gently with PBS pre-cooled at 4°C. The sample homogenate provided by the kit was added at a ratio of 100-200 ul per 1 million cells. After homogenisation or lysis, centrifuge at 10,000 g-12,000 g for 10 minutes and collect the supernatant for subsequent assays. Set up sample wells and blank control wells using a 96-well plate. Add the appropriate reagent solution to each well conformed to the manufacturer’s directions and mix well. The absorbance of SOD, GSH, MDA and CAT assays were recorded absorbance at 450nm, 340nm, 532nm and 520nm respectively using a microplate reader (Spark® 20M Multimode Microplate Reader, Tecan, Switzerland). All assays were replicated three times. The final calculation of enzyme activity in the samples was carried out.

2.3; 2.4; 2.6; 2.7 and 2.8 what were the controls in experiments? please add information (on Figures controls were placed)

Response: Sorry for our carelessness. In this study, the control cells treated with 1% DMSO. We have added this information in the revised manuscript.

Round 2

Reviewer 2 Report

Authors corrected manuscript according to comments and suggestions, now manuscript can be accepted for publication.